psychology

developmental psychology, moral self-licensing, sharing, cheating, morality

**Author for correspondence:**
Sophie Cameron
e-mail: sophie.cameron@uqconnect.edu.au

# Does helping now excuse cheating later? An investigation into moral balancing in children

Sophie Cameron[1], Matti Wilks[2] and Mark Nielsen[1,3]

[1]Early Cognitive Development Centre, School of Psychology, University of Queensland, Brisbane, Queensland, Australia
[2]Department of Psychology, Yale University, New Haven, CT 06520, USA
[3]Faculty of Humanities, University of Johannesburg, Johannesburg, South Africa

SC, 0000-0001-8378-0433; MN, 0000-0002-0402-8372

We often use our previous good behaviour to justify current immoral acts, and likewise perform good deeds to atone for previous immoral behaviour. These effects, known as moral self-licensing and moral cleansing (collectively, moral balancing), have yet to be observed in children. Thus, the aim in the current study was to investigate the developmental foundations of moral balancing. We examined whether children aged 4–5 years ($N = 96$) would be more likely to cheat on a task if they had previously helped a puppet at personal cost, and less likely to cheat if they had refused to help. This hypothesis was not supported, suggesting either that 4–5-year-old children do not engage in moral balancing or that the methodology used was not appropriate to capture this effect. We discuss implications and future research directions.

## 1. Introduction

Adults are motivated to maintain a positive moral self-image, but may also be tempted to engage in immoral behaviour [1]. Both can be achieved by engaging in moral balancing, in which the amount of moral and immoral deeds performed are balanced across time [2,3]. This involves two complementary effects: moral self-licensing and moral cleansing [4].

Moral self-licensing occurs when previous moral behaviour boosts one's perception of their moral self above a usual baseline. This may provide 'licence' for immoral acts to be performed, while not endangering self-perceptions as a moral person [5,6]. For example, someone who frequently donates to charity may feel less guilty for not giving change to a panhandler, as they

can refer to their previous history of charitable giving to assert that they are, on the whole, a generous person. Moral cleansing (or moral compensation) refers to situations when an immoral or antisocial act is followed by a moral or prosocial one [7,8]. Violating one's moral standards causes moral distress and a dip in an individual's self-worth, and thus, the positive act is performed to regain some of that lost self-worth [1]. Overall, this suggests that moral self-regulation is a dynamic process, in which moral acts may predict immoral ones, and conversely immoral acts may predict moral ones.

However, moral balancing in adults is far from completely understood, with some studies failing to find the effect at all (Blanken *et al*. [9]). Others have revealed both moral balancing and the opposite pattern, moral consistency, in which previous moral behaviour predicts future moral behaviour, depending on the situation (Conway & Peetz [10]; Gneezy *et al*. [11]; Joosten *et al*. [7]). In addition to the mixed findings in adults, there is scant research devoted to examining whether children engage in moral licensing. Considering the developmental origins of this behaviour may shed some light on the mixed findings in adults [12]. In particular, understanding when children begin to display this behaviour will allow us to examine what other cognitive developments are occurring at this time, which may point to relevant cognitive mechanisms that drive such behaviour.

There are many reasons to think that young children might morally license. Children show both antisocial and prosocial motivations early in development [13–19]. From as young as 22 months, children begin to show signs of complex moral emotions such as guilt and pride [20,21], emotions that are proposed to mark the beginning of the ability to appraise how actions align with one's own internal standards for morality [22]. These emotions occur even if children believe they are unobserved [20], suggesting that children at this age have a perception of themselves as moral beings outside of the parameters of adult punishment and reward.

The emergence of these self-referential moral emotions has been interpreted to suggest that children are starting to form at least a rudimentary moral identity—something that seems necessary in order to make the comparisons to the ideal moral-self thought to underlie moral balancing. Research later in development suggests that children have an explicit understanding of their moral identity by age 5. At this age, they will respond consistently to questions related to their moral character and preferences [20,23].

Children have also been shown to demonstrate consistency in their moral behaviour. Beginning around age 5, children increasingly begin to understand that trait descriptions of behaviour, such as 'you're a very helpful person' are enduring, and predictive of future behaviour [24,25]. Around the same time children begin to show moral consistency in their own behaviour, in a paradigm similar to the foot in the door effect [26–30], especially if their previous actions are linked back to a trait description [26,31]. Thus, we know from a young age, children have prosocial motivations and a developing moral identity. A preference for consistency suggests that children have a motivation to maintain a positive perception of their morality, much like adults.

However, to our knowledge, only one study has directly attempted to investigate the moral self-licensing effect in children. Tasimi & Young [32] asked children between the ages of 6 and 8 years old to recall a previous good, bad or neutral deed and then gave them the opportunity to behave prosocially towards a third party. Notably, they did not find an effect of licensing, but instead found that children engaged in moral consistency—those that recalled a good deed were more likely to behave prosocially than those who recalled a bad or neutral deed. This finding might suggest that children of this age do not engage in moral licensing. Alternatively, the study may have been unable to detect the effect. Notably, research with adults suggests that moral self-licensing is more likely to occur for recent, concrete behaviours [3,10]. As Tasimi & Young [32] had no control over the time or form of the behaviour recalled, variation in this regard may explain the lack of licensing effect.

The aim in the current study was thus to explore moral self-licensing in young children. Notably, we employed a behavioural task to manipulate the initial moral behaviour, allowing for a much higher degree of control. This meant that the domain of the moral behaviour and the time when it was performed was consistent for all children. This behavioural manipulation took the form of a series of games that the children played with a puppet, during which the child was sometimes able to provide help to the puppet. Children were then left alone to play a second game, with the opportunity to cheat in order to obtain a greater reward. If children engage in moral licensing, it was expected that children who had helped in the initial game would cheat to a greater degree. We further reasoned that if children display moral cleansing, children who did not help would cheat below the baseline rate.

# 2. Material and methods

## 2.1. Participants

A total of 145 children participated in this study. Of these, 23 were excluded from analysis—5 due to experimenter error, 10 due to not performing the moral behaviour expected in the initial task (further details below) and 8 who refused to be alone for the target task. An additional 13 children participated in a fourth condition, that was later dropped. This condition, which was included in the pre-registration on the Open Science Framework, was dropped due to high rates of attrition (38.46% of those tested in this condition had to be excluded). The final sample consisted of 96 children (49 females) between the ages of 48 and 71 months ($M = 59.84$, s.d. $= 7.42$). This sample size was decided *a priori* based on calculations by G*Power which specified 32 participants per cell to achieve 0.8 power to detect medium effect ($f = 0.3$) with $\alpha = 0.05$.[1] Demographics were not collected for this study, but generally families tested at this laboratory are primarily Oceanian (including Australia, New Zealand, Papua New Guinea and surrounding regions—71.07%). A large proportion of the remainder are of Asian heritage (12.86%), with the minority European (7.5%), American (2.5%) and African (1.4%). While the majority of the children only speak English, 21.42% also speak another language. Parents at this laboratory are typically highly educated, with the vast majority completing high school as well as further education, such as a trade, certification or bachelor's degree (92.86%).

## 2.2. Design

This was a between-participants design, in which children's moral behaviour in the initial phase of the experiment was the independent variable. The circumstances were manipulated so that children engaged in either no morally relevant behaviour (control), moral behaviour (moral) or immoral behaviour (immoral). Children were randomly assigned to one of these three conditions. After this initial task, children completed the target task, which assessed the amount that children cheated in a simple throwing game. The frequency and severity of cheating were the dependent variables.

## 2.3. Procedure

All participants were tested in a dedicated child-friendly testing laboratory and parents provided consent for participation. The methods for this study were pre-registered; the registration is available at https://osf.io/7re83/.

### 2.3.1. Initial task

The first phase of the experiment was designed to induce moral behaviour in the children, using tasks adapted from Green *et al.* [33]. Children were told they were going to play a series of games alongside a puppet and would be able to win some stickers. To increase motivation, before the beginning of the games, the children were shown an assortment of stickers and asked to select their favourite three. The experimenter told them they would have the chance to win the stickers, and then they began the games. There were three different games (detailed below), which were each played with a different puppet. The order of the games and the puppet which played each game was consistent across children.

Before each game, the experimenter introduced the child to the puppet, and explained that if the child finished the game before the time was up, they could win one of the stickers they had selected, and if the puppet finished the game, each would win a sticker. It was emphasized that if both the child and the puppet finished the game, each would win a sticker. The experimenter controlled the puppet so that it finished the game slightly after the child. Once the child and puppet were finished the experimenter said 'Stop, time's up', and gave the child their sticker.

*Marble Game:* The marble game consisted of a plank of wood with nine circular divets in it. The child had nine marbles and had to place a marble in each of the divets in the plank.

*Rod Game:* The rod game consisted of a plank of wood with four different coloured rods sticking vertically out of it. The child had nine wooden blocks of each colour and had to place these blocks on the matching colour rods.

---

[1]The pre-registration includes a sample size of 25 per cell, this was based on a power analysis that we later determined to be inaccurate.

*Puzzle Game:* The puzzle game consisted of a brightly coloured children's puzzle. It had nine pieces featuring aquatic animals, each with a corresponding place in the puzzle. The child's task was to put the pieces in the matching spot on the puzzle.

### 2.3.2. Moral manipulation

The circumstances of the initial task were manipulated to induce morally relevant behaviour in the two experimental conditions. In the control condition, the puppet did not require any help, and this served as a baseline. In the moral condition, the puppet required help and the children provided it, thus performing a moral behaviour. Conversely in the immoral condition, the children did not provide help to the puppet, thereby behaving immorally.[2]

*Moral:* On the last game, the puppet was missing the last two pieces required to complete the puzzle. When the child had finished their game, the puppet realized these pieces were missing and expressed distress in three, scripted prompts increasing in emotional affect, adapted from Green *et al.* [33]. In this condition, the child had two extra pieces, and thus could help the puppet without sacrificing their own reward. Based on previous research, we expected most children to help [33], and indeed 94.11% did. Once the children helped the puppet, the puppet thanked them and completed the game. The experimenter then said the time was up and as both games were complete, both the child and puppet received a sticker. If the child did not help, the experiment continued but the child was excluded from analysis.

*Immoral:* As above in the moral condition, however, the children did not have any extra pieces to help the puppet. This meant that helping would require them to give the puppet pieces from their own game and sacrifice their own reward. As, such we did not expect children to help, and indeed 80% did not. Once the puppet had completed all three distress prompts the experimenter said that the time was up. As the child had completed their game, they received a sticker, but the puppet did not. If the child did help, the experiment continued but the child was excluded from analysis.

### 2.3.3. Target task

For the target task, children played a game that involved standing behind a line and throwing balls into two baskets. One basket was placed close to the line, and the children won one sticker for every ball thrown into it. The other basket was placed further away from the line and the child won three stickers for every ball they got in it. All children were provided with 20 balls to play this game. The reward for each basket was first explained to the children. They were then asked about how many stickers each basket was worth. If they answered incorrectly, the rules were repeated and they were asked again. This was repeated until they were able to recall correctly. No child required more than three explanations to understand the rules.

The experimenter then explained 'the most important rule of the game', which was that in order to throw the balls, the child had to stand behind a line marked on the floor, and that crossing the line was cheating. The child was then given the opportunity to practice throwing in each of the baskets, and the number of stickers they were worth was repeated. Finally, the experimenter explained that if they missed a bucket, they were permitted to cross the line to retrieve the ball, but they must return behind the line to throw the ball again.

Once the rules were explained, the experimenter said that they and the child's parent had to do work in a nearby room, and the child would be playing alone. The experimenter and parent then left the room and observed through a one-way mirror. They returned either after 5 min, when the child ran out of balls, or when the child knocked on the door to indicate they wished to stop playing. The experimenter gave the children the appropriate number of stickers, and then asked the child to recall the rules of the game. If the child said they did not recall, they further prompted by asking where the child had to stand to throw the ball. All children recalled this rule at this time.[3] After completing the task, children and parents were debriefed and the child was offered a prize as a thank you for participating.

---

[2]It could be argued that failing to perform a moral behaviour is not necessarily immoral. However, moral cleansing literature requires only that the act threaten one's sense of morality. We believe that withholding help from a distressed puppet should trigger this. For clarity, we refer to this behaviour as immoral.

[3]Before the end of the study, the experimenter also asked the child if they had cheated, which was later coded for lying. However, this measure was dropped due to a lack of variability (only one child admitted to cheating).

## 2.4. Data coding

Children's behaviour in the target task was coded to produce three separate measures of cheating: (i) total times the child cheated, (ii) the severity of cheating, and (iii) cheating latency. A sample of 10.41% of the videos were coded by a second rater, blind to experimental condition, for inter-rater reliability.

### 2.4.1. Total cheating

This variable represented the total number of times the child cheated. Any instance of at least part of the child's foot being over the line when a ball was thrown was counted and weighted equally. All instances of cheating were summed together to create the final score. Inter-rater reliability for this measure was very high; the average ICC was 0.96, 95% CI [0.84, 0.99].

### 2.4.2. Cheating severity

This variable quantified differences in the severity of children's cheating. Different instances of cheating were weighted differently according to their severity. Cheating that involved just part of the child's feet being over the line was coded as a 1. Cheating that involved an entire foot being over the line was coded as a 2. Cheating that involved the entirety of both feet being across the line, but not beyond the first bucket was coded as a 3. Cheating that involved the child being beyond the first bucket was coded as a 4. These values were summed together, and then divided by the total number of throwing attempts the child made, in order to account for variation in the frequency of throwing. Inter-rater reliability for this measure was very high; the average ICC was 0.99, 95% CI [0.98, greater than 0.99].

### 2.4.3. Cheating latency

This variable represented how long it took the children to start cheating. It was coded as the amount of time elapsed (in seconds) after the experimenter left the room that the first instance of cheating took place. Inter-rater reliability for this measure was very high; the average ICC was 0.94, 95% CI [0.72, 0.99].

# 3. Results

## 3.1. Preliminary analysis

Preliminary analyses revealed no effect of age or sex on any of the dependent variables ($p$s > 0.253). As such, these variables were excluded from future analysis.

## 3.2. Main analysis

Of the 96 children tested, 82.3% cheated at least once, with the average instances of cheating being 12.02. Of those who cheated, the average delay before cheating was 28.23 s.

### 3.2.1. Total cheating

A univariate analysis of variance revealed no difference in total instances of cheating between the control ($M = 13.31$, s.d. = 11.78), moral ($M = 11.38$, s.d. = 12.51) and immoral ($M = 11.38$, s.d. = 11.74), conditions ($F_{2,96} = 0.28$, $p = 0.758$, $\eta^2 < 0.01$, 95% CI [0.00, 0.04]). Bayesian analysis confirmed that there was substantial evidence against the alternative hypothesis ($BF_{01} = 0.12$).

### 3.2.2. Cheating severity

A univariate analysis of variance revealed no difference in the severity of the cheating in the control ($M = 1.03$, s.d. = 0.86), moral ($M = 0.71$, s.d. = 0.91) and immoral ($M = 0.88$, s.d. = 0.79), conditions ($F_{2,96} = 1.13$, $p = 0.327$, $\eta^2 = 0.02$, 95% CI [0.00, 0.08]). Bayesian analysis confirmed that there was substantial evidence against the alternative hypothesis ($BF_{01} = 0.24$).

### 3.2.3. Cheating latency

A univariate analysis of variance revealed no difference between the onset of cheating between the control ($M = 29.62$, s.d. $= 46.67$), moral ($M = 30.56$, s.d. $= 39.89$) and immoral ($M = 24.59$, s.d. $= 39.72$), conditions ($F_{2,81} = 0.15$, $p = 0.859$, $\eta^2 < 0.01$, 95% CI [0.00, 0.03]). Bayesian analysis confirmed that there was substantial evidence against the alternative hypothesis ($BF_{01} = 0.12$).

## 4. Discussion

In this study, we explored whether children aged 4–5 years would engage in moral balancing. Specially, we investigated whether varying children's helping behaviour on an initial task would affect their subsequent levels of cheating. If children did show moral self-licensing, those who helped could be expected to show highest levels of cheating. If children showed moral cleansing, then we expected that those who did not help would show the lowest levels of cheating. Surprisingly, we found no differences between children who helped, did not help and a neutral baseline on all measures of cheating.

This null result can be interpreted in two different ways. The first is that children in the age range selected (4–5 years old) do not show the moral balancing effect. As discussed, this age range was selected as it is a time of considerable moral development, including the presence of self-referential emotions [20,21,23] and moral consistency effects, suggesting the formation of a moral identity [26–30]. However, it is possible that moral balancing only emerges later, as children undergo considerable moral development as they progress through middle childhood and adolescence. Children aged 4–5 years may have a rudimentary understanding of moral identity, but it may not be complex enough to produce the dynamic effects predicted by moral balancing. In particular, research suggests that from early childhood to adolescence, children develop a stronger sense of moral motivation and ownership over their own actions, although it is unclear at what age this shift occurs [34,35]. Therefore, children may not show moral balancing until later in development.

Alternatively, it is possible that children in this age range have the potential to engage in moral balancing, but the current methodology was not appropriate to capture it. Notably, we found large individual differences in cheating behaviour which may have washed out condition effects. This could be a result of the task itself—the 5 min time allocated was extensive and the cheating was relatively low stakes, thus children may have been more inclined to cheat than if they had been in a room with others. Additionally, we used puppets as social partners, which may yield different outcomes than if children helped their peers [36,37].

In addition to this, it is possible that our moral manipulation, while reliable, did not sufficiently produce differences in children's moral identities. It is suggested that moral balancing works by causing one's sense of moral identity to shift above or below desired baseline, prompting moral or immoral behaviour accordingly [4]. Based on this interpretation, we would need to produce both a change in behaviour and moral identity. While we did not measure this explicitly, it is possible that our manipulation did not prompt these changes in moral identity. Anecdotally, children who helped often did so with very little hesitation and commented that they seemed to have the puppet's puzzle pieces by mistake. Although the puppet thanked them for helping, it is possible that the children themselves did not see this as a helping action, so much as fixing a mistake made by the experimenter. Perhaps this act was not costly enough for children to recognize it as a moral act and create a change in their perceptions of their moral identity. Furthermore, children in the immoral condition may have felt that sacrificing their own reward for the puppet was too costly, and, therefore, did not experience any shift in moral identity from not helping. It would be beneficial for future research to employ a check to ensure that children perceive their behaviour in the moral manipulation as morally relevant.

While we have not identified the age in which moral licensing occurs, this study prompts several directions for future research. In particular, it speaks to the need to employ salient moral manipulations, as well as using an outcome measure that has been shown to be easily manipulated. Finally, employing a broader age range would shed light on whether these results are a product of the experimental design, or do genuinely reflect a lack of moral balancing in young children. With this knowledge, we can begin to understand the development of the mechanisms that underpin our tendency to engage in moral balancing, as well as other forms of moral self-regulation.

Ethics. This study received ethics approval from the university board (approval number 2018000918) and informed consent was obtained from every child's guardian before testing commenced.

Data accessibility. The data associated with this manuscript can be found at the Open Science Framework, doi:10.17605/OSF.IO/7RE83 (https://osf.io/7re83/).

Authors' contributions. S.C. and M.N. collaborated on the study design. S.C. collected and analysed the data, with input from M.N. and M.W. All authors collaborated on the drafting and reviewing of the manuscript.

Competing interests. We have no competing interests to declare.

Funding. No outside funding was used for this project.

Acknowledgements. The authors would like to thank all the participating children and their families for their support during data collection.

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
