## [Peer Review File · Royal Society Open Science]

Review History

RSOS-202296.R0 (Original submission)

Review form: Reviewer 1 (Ben Kenward)

Is the manuscript scientifically sound in its present form?

No

Are the interpretations and conclusions justified by the results?

No

Is the language acceptable?

Yes

Do you have any ethical concerns with this paper?

No

Have you any concerns about statistical analyses in this paper?

Yes

Recommendation?

Accept with minor revision (please list in comments)

Comments to the Author(s)

There is sometimes a tendency to argue that single-experiment studies with null-results are not worth publishing. On the other hand, I tend to agree with the authors that there are a number of things to be learned from this study, even if they are primarily methodological. Given that even medium-sized behaviour samples from young children can be hard to come by, I think this study does indeed make a publishable contribution. Here are some specific comments:

1. I think the abstract should be brought more into line with the discussion. Whereas the discussion appropriately concludes that the null result could be due to methodological issues, and contains some discussion of those issues that might well be the most useful contribution of this work, the abstract does not accurately reflect this. Rather, it claims the “[the moral licensing] hypothesis was not supported, suggesting that 4-5-year-old children do not engage in moral self-licensing.” The abstract should reflect the fact that, as discussed, the results might reflect methodological issues rather than anything else, and therefore should make weaker claims about moral licensing in children of this age.

2. For interpretation of null results, it’s important to present confidence intervals on effect sizes. This is more valuable than a power analysis, which tells you what you should have guessed before you had any data. Now you do have data, so you can present confidence intervals on Cohen’s *d* so that we can see how likely it is that a small or medium effect in this paradigm might exist undetected.

3. On the CogDevSoc email list (which I allow myself a reference to because I have seen the last author post there around a year ago) there has recently been a provocative debate about the extent to which we can learn things about real behaviour from observing children interact with puppets, when children are old enough to know puppets are not real. The methodological paradigm used here does seem to be suited to some further reflection on this topic, because the manipulation assumes that children experience thoughts that are in some way morally real about their interactions with a puppet. The authors show some acknowledgement of that issue (“While the puppet did thank them for helping, it is possible that the children themselves did not see this as a helping action, so much as fixing a mistake made by the experimenter”) but I think they need to give it a little more attention.

4. What about the possibility that some children experience moral licensing and other moral cleansing, so that on a group level it cancels out? E.g. I’m thinking of a recent meta-analysis of positive and negative spill-over in the context of pro-environmental behaviour (Maki et al. 2019, <https://www.nature.com/articles/s41893-019-0263-9>), where they find that the overall estimate of the effect is close to zero because sometimes you get positive spill-over and sometimes negative. In future work, some measurement or manipulation of a relevant factor (e.g. the role reinforcement vs. positive affect aspects of positive behaviour) could be included to try and reveal this.

5. I was curious about the ethnicity of lab participants being described as “Oceanian” and looking it up didn’t make me much the wiser. Perhaps it would help (as the Oceanian geographical area is huge) to state the location of the lab.

6. There is a spurious capital I in section 3.2.1

Review form: Reviewer 2

Is the manuscript scientifically sound in its present form?

Yes

Are the interpretations and conclusions justified by the results?

Yes

Is the language acceptable?

Yes

Do you have any ethical concerns with this paper?

No

Have you any concerns about statistical analyses in this paper?

No

Recommendation?

Accept with minor revision (please list in comments)

Comments to the Author(s)

The current study investigated whether an experimental manipulation – inducing moral behavior, immoral behaviour, or a control condition – causes changes in subsequent cheating behaviors with a sample of 4- to 5-year-olds. The premise was that the idea of moral licensing would be supported if those who previously behaved morally were more likely to cheat. The data did not support this premise. Overall, I found the idea interesting. The study was not without its blemishes, but frankly I found the upfront and thorough reporting of all such information refreshing – too many researchers still try to sweep important information under the rug and pray a reviewer doesn't ask about it... Here, just about every bit of data I would've liked to have seen was reported. And I found the reasons for the decisions made (e.g., exclusions) justifiable. So, I appreciated the transparent, open science approach to the study.

I have some comments and points about the manuscript itself that I think would benefit the paper.

First, the interpretation of the immoral manipulation is a little questionable. The authors note: "moral cleansing literature requires only that the act threaten one's sense of morality. We believe that withholding help from a distressed puppet should trigger this." In the immoral condition, children didn't have extra pieces to give to the puppet other than their own which would require them to sacrifice their own reward. It seems like a stretch to consider it immoral to not disadvantage yourself in this type of game. As a comparison, studies using sharing/donating tasks consider a 50:50 split to be a "moral" outcome. Even among older children, you don't often see children allot more than half of a desired resource to another child in those types of tasks. Therefore, I find it unlikely that the immoral condition would threaten anyone's sense of morality. There may have been a stronger argument here the reward was given on a continuum (e.g., 1 sticker per correctly placed block) and children could've had the opportunity to sacrifice something to the puppet in that game without giving up their entire reward. Nevertheless, I think it's prudent to mention this is a limitation.

Second, the cheating task seems to have yielded very high rates of cheating, perhaps because there was very little harm done by cheating. If the cheating was higher stakes, for instance, with the child's performance being compared to someone else's and winner being whoever had more

points, then children may have been more selective in their cheating. This should be noted as a limitation and/or potential future direction.

Third, the introduction seems to set the stage for moral balancing vs. moral consistency. It appears these findings support the moral consistency interpretation, but that is hardly followed up on in the discussion. If my interpretation of the findings is correct, it would make more sense to weave this bit about competing hypothesis throughout the paper and clarify which predictions would be expected based on each account of children's behavior.

Fourth, I found the part about moral cleansing somewhat detracting here. This concept isn't directly tested in this study and therefore it is unnecessary to describe. Instead, I think it may be more fruitful to review some of the age-related findings around other elements of children's self and preferences, whether that be (moral) values or how identity/character-based praise impacts behavior (e.g., some of Joan Grusec's studies).

Fifth, some greater discussion about the implications of these findings would be helpful. These results mean something other than a failure to identify moral licensing in early childhood. I would be interested in hearing how these findings connect to a broader literature about the moral self and behavior.

Lastly, below are a handful of typos and minor edits:

-Line 36: colon not semicolon.

-Typo in citation 20

-“From as young as 22 months, children begin to show complex moral emotions such as guilt and pride (17,18)”

- Precursors to – or at best “signs of”

-P3, L11“Around the same time children begin to show moral consistency in their own behavior, in a paradigm similar to the foot in the door effect (23–27)”

- Unclear how foot in the door effect relates to moral consistency here – is this necessary to mention?

-“While the majority of the children only speak English,”

- Probably want the past tense throughout the method

-P5, l39: “no difference In total”

Decision letter (RSOS-202296.R0)

Dear Ms Cameron

On behalf of the Editors, we are pleased to inform you that your Manuscript RSOS-202296 "Does helping now excuse cheating later? An investigation into moral self-licensing in children" has been accepted for publication in Royal Society Open Science subject to minor revision in

accordance with the referees' reports. Please find the referees' comments along with any feedback from the Editors below my signature.

Please submit your revised manuscript and required files (see below) no later than 7 days from today's (ie 28-Apr-2021) date. Note: the ScholarOne system will 'lock' if submission of the revision is attempted 7 or more days after the deadline. If you do not think you will be able to meet this deadline please contact the editorial office immediately.

on behalf of Dr Teodora Gliga (Associate Editor) and Essi Viding (Subject Editor)
openscience@royalsociety.org

Associate Editor Comments to Author (Dr Teodora Gliga):

Associate Editor: 1

Comments to the Author:

I have now received two reviews and, as you will see, they are both positive about your manuscript yet they both require further clarifications and discussion. Please provide a detailed response to these queries and modify the manuscript accordingly. In particular, make sure the abstract reflects all aspects of the study (you investigated both licensing and cleansing) and that all manipulations and terminology is well justified (i agree that not giving away one's toys may not fit with our understanding of immoral yet it is likely that children in this age range know that it is not nice not to help; would pro-social and anti social behavior work better?). Please also provide Bayesian statistics to test for support for the null hypothesis.

Associate Editor: 2

Comments to the Author:

(There are no comments.)

Reviewer comments to Author:

Reviewer: 1

Comments to the Author(s)

There is sometimes a tendency to argue that single-experiment studies with null-results are not worth publishing. On the other hand, I tend to agree with the authors that there are a number of things to be learned from this study, even if they are primarily methodological. Given that even

medium-sized behaviour samples from young children can be hard to come by, I think this study does indeed make a publishable contribution. Here are some specific comments:

1. I think the abstract should be brought more into line with the discussion. Whereas the discussion appropriately concludes that the null result could be due to methodological issues, and contains some discussion of those issues that might well be the most useful contribution of this work, the abstract does not accurately reflect this. Rather, it claims the “[the moral licensing] hypothesis was not supported, suggesting that 4-5-year-old children do not engage in moral self-licensing.” The abstract should reflect the fact that, as discussed, the results might reflect methodological issues rather than anything else, and therefore should make weaker claims about moral licensing in children of this age.
2. For interpretation of null results, it’s important to present confidence intervals on effect sizes. This is more valuable than a power analysis, which tells you what you should have guessed before you had any data. Now you do have data, so you can present confidence intervals on Cohen’s *d* so that we can see how likely it is that a small or medium effect in this paradigm might exist undetected.
3. On the CogDevSoc email list (which I allow myself a reference to because I have seen the last author post there around a year ago) there has recently been a provocative debate about the extent to which we can learn things about real behaviour from observing children interact with puppets, when children are old enough to know puppets are not real. The methodological paradigm used here does seem to be suited to some further reflection on this topic, because the manipulation assumes that children experience thoughts that are in some way morally real about their interactions with a puppet. The authors show some acknowledgement of that issue (“While the puppet did thank them for helping, it is possible that the children themselves did not see this as a helping action, so much as fixing a mistake made by the experimenter”) but I think they need to give it a little more attention.
4. What about the possibility that some children experience moral licensing and other moral cleansing, so that on a group level it cancels out? E.g. I’m thinking of a recent meta-analysis of positive and negative spill-over in the context of pro-environmental behaviour (Maki et al. 2019, <https://www.nature.com/articles/s41893-019-0263-9>), where they find that the overall estimate of the effect is close to zero because sometimes you get positive spill-over and sometimes negative. In future work, some measurement or manipulation of a relevant factor (e.g. the role reinforcement vs. positive affect aspects of positive behaviour) could be included to try and reveal this.
5. I was curious about the ethnicity of lab participants being described as “Oceanian” and looking it up didn’t make me much the wiser. Perhaps it would help (as the Oceanian geographical area is huge) to state the location of the lab.
6. There is a spurious capital I in section 3.2.1

Reviewer: 2

Comments to the Author(s)

The current study investigated whether an experimental manipulation – inducing moral behavior, immoral behaviour, or a control condition – causes changes in subsequent cheating behaviors with a sample of 4- to 5-year-olds. The premise was that the idea of moral licensing would be supported if those who previously behaved morally were more likely to cheat. The data did not support this premise. Overall, I found the idea interesting. The study was not without its blemishes, but frankly I found the upfront and thorough reporting of all such information refreshing – too many researchers still try to sweep important information under the rug and

pray a reviewer doesn't ask about it... Here, just about every bit of data I would've liked to have seen was reported. And I found the reasons for the decisions made (e.g., exclusions) justifiable. So, I appreciated the transparent, open science approach to the study.

I have some comments and points about the manuscript itself that I think would benefit the paper.

First, the interpretation of the immoral manipulation is a little questionable. The authors note: "moral cleansing literature requires only that the act threaten one's sense of morality. We believe that withholding help from a distressed puppet should trigger this." In the immoral condition, children didn't have extra pieces to give to the puppet other than their own which would require them to sacrifice their own reward. It seems like a stretch to consider it immoral to not disadvantage yourself in this type of game. As a comparison, studies using sharing/donating tasks consider a 50:50 split to be a "moral" outcome. Even among older children, you don't often see children allot more than half of a desired resource to another child in those types of tasks. Therefore, I find it unlikely that the immoral condition would threaten anyone's sense of morality. There may have been a stronger argument here the reward was given on a continuum (e.g., 1 sticker per correctly placed block) and children could've had the opportunity to sacrifice something to the puppet in that game without giving up their entire reward. Nevertheless, I think it's prudent to mention this is a limitation.

Second, the cheating task seems to have yielded very high rates of cheating, perhaps because there was very little harm done by cheating. If the cheating was higher stakes, for instance, with the child's performance being compared to someone else's and winner being whoever had more points, then children may have been more selective in their cheating. This should be noted as a limitation and/or potential future direction.

Third, the introduction seems to set the stage for moral balancing vs. moral consistency. It appears these findings support the moral consistency interpretation, but that is hardly followed up on in the discussion. If my interpretation of the findings is correct, it would make more sense to weave this bit about competing hypothesis throughout the paper and clarify which predictions would be expected based on each account of children's behavior.

Fourth, I found the part about moral cleansing somewhat detracting here. This concept isn't directly tested in this study and therefore it is unnecessary to describe. Instead, I think it may be more fruitful to review some of the age-related findings around other elements of children's self and preferences, whether that be (moral) values or how identity/character-based praise impacts behavior (e.g., some of Joan Grusec's studies).

Fifth, some greater discussion about the implications of these findings would be helpful. These results mean something other than a failure to identify moral licensing in early childhood. I would be interested in hearing how these findings connect to a broader literature about the moral self and behavior.

Lastly, below are a handful of typos and minor edits:

-Line 36: colon not semicolon.

-Typo in citation 20

-"From as young as 22 months, children begin to show complex moral emotions such as guilt and pride (17,18)"

• Precursors to – or at best "signs of"

-P3, L11 “Around the same time children begin to show moral consistency in their own behavior, in a paradigm similar to the foot in the door effect (23–27)”

- Unclear how foot in the door effect relates to moral consistency here – is this necessary to mention?

-“While the majority of the children only speak English,”

- Probably want the past tense throughout the method

-P5, l39: “no difference In total”

===PREPARING YOUR MANUSCRIPT===

===PREPARING YOUR REVISION IN SCHOLARONE===

Author's Response to Decision Letter for (RSOS-202296.R0)

See Appendix A.

Decision letter (RSOS-202296.R1)

Dear Ms Cameron,

It is a pleasure to accept your manuscript entitled "Does helping now excuse cheating later? An investigation into moral balancing in children" in its current form for publication in Royal Society Open Science.

on behalf of Dr Teodora Gliga (Associate Editor) and Essi Viding (Subject Editor)
openscience@royalsociety.org

Appendix A

Dear Dr. Theodora Giulga and Dr. Essi Viding,

Thank you for the acceptance of my manuscript into Royal Society Open Science. I found the reviewer comments very helpful and have revised the manuscript accordingly.

In your acceptance, you highlighted a few of your own concerns, in addition to those of the reviewers. Specifically, you noted that:

1. The abstract focuses on moral self-licensing, while the main paper discussed both moral self-licensing and moral cleansing.

I agree that the abstract as it was may have given a misleading impression of the study. To rectify this, I have changed the abstract to discuss moral balancing, incorporating both moral self-licensing and moral cleansing as follows:

We often use our previous good behavior to justify current immoral acts, and likewise perform good deeds to atone for previous immoral behaviour. These effects, known as moral self-licensing and moral cleansing (collectively, moral balancing), have yet to be observed in children. Thus the aim in the current study was to investigate the developmental foundations of moral balancing. We examined whether children aged 4-5 years ($N = 96$) would be more likely to cheat on a task if they had previously helped a puppet at personal cost, and less likely to cheat if they had refused to help. This hypothesis was not supported, suggesting either that 4-5-year-old children do not engage in moral balancing, or that the methodology used was not appropriate to capture this effect. We discuss implications and future research directions.

I have also edited the title from “An investigation into moral self-licensing in children” to “An investigation into moral balancing in children”.

2. You also requested that we make sure that all terminology and manipulations are justified. In particular, you expressed concern that the immoral manipulation may not truly count as immoral, and that the terms anti-social and pro-social behaviour may be more appropriate.

I understand querying the immoral manipulation and have discussed this in my response to reviewers. I also understand the motivation to change the terminology to pro-social and anti-social. However, given that this study investigates the effect known as moral balancing, I am concerned that changing the terminology away from moral / immoral may unnecessarily confuse readers and hence prefer to leave this aspect of the manuscript unchanged.

3. You requested Bayesian statistics to support the null result.

For each test conducted, I have included the Bayes factor for the result.

For the remainder of the reviewer comments, I have considered the concerns and revised the manuscript accordingly. I look forward to hearing from you again.

Kind regards,

Sophie Cameron

Response to Reviewers

Below I include a response to each of the comments made by the reviewers. Points are ordered according to how they appear in the reviews.

Response to Reviewer 1.

R1.1: Reviewer 1 thought that the abstract should be brought more in line with the discussion, specifically to reflect that the null result could be due to methodological limitations, and not conclusive evidence of a lack of moral balancing at this age.

I agree that the original wording of the abstract made too strong a claim about the interpretation of this null effect. I have edited it to be more conservative, specifically changing the last two sentences from:

This hypothesis was not supported, suggesting either that 4-5-year-old children do not engage in moral self-licensing. We discuss implications and future research directions.

To:

This hypothesis was not supported, suggesting either that 4-5-year-old children do not engage in moral *balancing*, or that *the methodology used was not appropriate to capture this effect*. We discuss implications and future research directions.

R1.2: The reviewer stated that it was important to provide confidence intervals on effect sizes for the null effects, in order to determine the likelihood that a small or medium effect was undetected in this paradigm.

I was not aware of this convention, and so apologise for not including this calculation originally. For each ANOVA I have computed the eta squared effect size, which to the best of my knowledge is the appropriate effect size for an omnibus test. For each effect size I have also provided the confidence intervals.

R1.3: The reviewer referred to the 'Theory of Puppets' debate, which has been recently occurring in the developmental psychology community. Specifically, they expressed concerns that the children are aware that puppets are not real, and therefore their interactions with them may not be morally relevant.

I am aware of this debate, and certainly see its relevance to this study. The choice of a puppet as the child's partner in the games used was made for logistic reasons. I was concerned that children of the age range tested might be intimidated by the presence of another adult, and would not interact with them normally. Alternatively, using another child as the partner would require the child to act in the same way

that the puppet did (i.e., reading the script, getting increasingly upset), and I felt that this was also not realistic. Therefore, we used a puppet for the sake of an agent that could be controlled by the experimenter, and the children could treat as a peer.

I nonetheless agree that it is possible the children in this study were aware that the puppet was not real, and therefore did not see their interactions with it as morally relevant. I have included this as a limitation in the discussion, with some references to some papers that cover this debate in more depth:

Additionally, we utilized puppets as social partners, which may yield different outcomes than if children helped their peers (34,35).

34. Kominsky J, Lucca K, Thomas A, Frank M, Hamlin K. Simplicity and validity in infant research. 2020.

35. Revencu B, Csibra G. Opening the black box of early depiction interpretation: From whether to how in the Theory-of-Puppets debate. 2020.

R.1.4: The reviewer raised the possibility that there may be individual differences in children's proclivity to engage in moral consistency or moral balancing, such that the moral manipulation may have triggered different effects in different children. This would have resulted in a cancellation of the overall effect on the group level, leading to the null effect observed. They referred to a meta-analysis of moral self-licensing in environmental behaviour that had this occur (Maki et al, 2019).

I think that this is a very interesting point to raise, and I certainly believe that it is possible that some manipulations trigger moral licensing and some trigger moral consistency, resulting in a cancelling out of effect sizes when looking across studies (as in the meta-analysis described). However, given that the manipulation in this study was consistent across children, I think this explanation unlikely in the current paradigm. It is possible that some children's moral identities may be affected differently by the manipulation used (potentially due to different perception of the self as a 'helpful' individual), resulting in moral consistency or licensing effects that vary between children. However, this relies on a relatively complex understanding of the moral self in children of this age range and seems less parsimonious than methodological limitations explaining the null result. I have therefore not made any alterations to the manuscript regarding this.

R.1.5: Reviewer 1 said that the use of the term 'Oceanian' as a description of ethnicity was confusing, and asked for clarification.

I have edited this section of the methods to make it clearer which countries Oceanian refers to. Specifically, I have changed it from:

Demographics were not collected for this study, but generally families tested at this lab are primarily Oceanian (71.07%).

To:

Demographics were not collected for this study, but generally families tested at this lab are primarily Oceanian (including Australia, New Zealand, Papua New Guinea, and surrounding regions -71.07%).

R.1.6: The reviewer pointed out a spurious capital I in section 3.2.1

This has been corrected.

Response to Reviewer 2.

R.2.1: Reviewer 2 expressed concern that the immoral manipulation used did not truly count as immoral behaviour. They argue that the manipulation required the children to sacrifice all of their own reward in order to not be considered immoral and contrasted this with the common standard of a 50/50 split being considered moral in sharing / fairness paradigms. Therefore, they reasoned that children would not feel that they were being immoral by not giving up their own resources, and their moral identity would not be threatened in the sense required to trigger moral cleansing.

I agree that giving up your own resources for the sake of another is a self-less act, and not doing so may not be considered truly immoral. As stated in the method, we believe that the puppet's emotional reaction may have been enough to cause the children to feel some guilt over their lack of action. However, I still think it is a valid point that merits discussion as a limitation. As such, I have included the following (p. 7):

Furthermore, children in the immoral condition may have felt that sacrificing their own reward for the puppet was too costly, and therefore did not experience any shift in moral identity from not helping. It would be beneficial for future research to employ a check to ensure that children perceive their behavior in the moral manipulation as morally relevant.

R.2.2: The reviewer commented on the high rates of cheating in the main measure and suggested that this could be due to the fact that there was very little harm done by cheating. They suggested that if the situation had higher stakes, for example in competition with another child, there may have been less variation in cheating.

I think that the high rates of cheating are an important limitation of this study, and indeed have addressed this in the discussion. However, I had not considered the possibility that the low stakes of this task may have been contributing to the variation. To this end, I have edited the discussion to reflect this. Specifically:

Notably, we found large individual differences in cheating behavior which may have washed out condition effects. This could be a result of the task itself—the 5-minute time allocated was extensive and resulted in some children engaging in large amounts of cheating.

Has been edited to (p. 7):

Notably, we found large individual differences in cheating behavior which may have washed out condition effects. This could be a result of the task itself—the 5-minute time allocated was extensive and the cheating was relatively low stakes, thus children may have been more inclined to cheat than if they had been in a room with others.

R.2.3: Reviewer 2 felt that the introduction established this study as an investigation into the competing predictions of moral consistency and moral balancing. They feel that this is not followed up adequately in the discussion.

My intention when including the discussion of moral consistency in the introduction was to establish children's moral capabilities at this age, particularly with regards to the moral self. I did not intend to test for moral consistency. I also note that in this paradigm, moral consistency would predict significantly lower rates of cheating after helping the puppet (and higher rates after not helping), and therefore the current null result does not support the moral consistency perspective.

However, I do appreciate the comment that it is confusing to incorporate this literature in the introduction and then not follow up on it in the discussion. To rectify this, I referred to it in the discussion. Specifically:

This null result can be interpreted in two different ways. The first is that children in the age range selected (4-5 years old) do not show the moral balancing effect. As discussed, this age range was selected as it is a time of considerable moral development, including the presence of self-referential emotions (17,18,20). However, it is possible that moral balancing only emerges later, as children undergo considerable moral development as they progress through middle childhood and adolescence.

Now reads (p. 7):

This null result can be interpreted in two different ways. The first is that children in the age range selected (4-5 years old) do not show the moral balancing effect. As discussed, this age range was selected as it is a time of considerable moral development, including the presence of self-referential emotions (17,18,20) and moral consistency effects, suggesting the formation of a moral identity (23–27). However, it is possible that moral balancing only emerges later, as children undergo considerable moral development as they progress through middle childhood and adolescence.

R.2.4: The reviewer commented that the section on moral cleansing detracts from the main paper, as it isn't directly tested in the study.

I understand the reviewers concern, particularly given the previously discussed comment about the limitations of the immoral condition that was intended to test moral cleansing. Given this limitation, it is valid to argue that we did not adequately test the phenomenon of moral cleansing. However, we nevertheless intended to do so. Removing the discussion of moral cleansing from the paper leaves very little motivation for the immoral condition and may thus unnecessarily confuse the reader. My preference is thus to leave this section in.

R.2.5: Reviewer 2 felt that the discussion would benefit from some broader discussion of the implications of the findings, and that the results reflect more than a lack of this effect in early childhood.

I appreciate the reviewer's confidence in these results and their ability to add to the broader literature on the moral self and behaviour. However, I am personally hesitant to draw broader implications from the current findings. This study did not find evidence for moral balancing in children, although it did present some useful avenues for future investigation into this topic. It feels disingenuous to make further conclusions about the implications of the absence of moral balancing in this age range without further evidence of its absence.

R.2.6-7: The reviewer pointed out several typos

These have been corrected.

R.2.8: The reviewer suggested that the claim that: “From as young as 22 months, children begin to show complex moral emotions such as guilt and pride (17,18)” was too strong.

This has been edited to be more tentative. Specifically, it now reads:

From as young as 22 months, children begin to show signs of complex moral emotions such as guilt and pride (17,18)

R.2.9: The reviewer expressed confusion over the relevance of the foot in the door effect in the sentence: “Around the same time children begin to show moral consistency in their own behavior, in a paradigm similar to the foot in the door effect (23–27)”

The was included to give a brief description of the methodology of the studies described – in particular that they involve prompting participants to continue agreeing to increasingly costly requests out of a desire to remain personally consistent. I believe this is a succinct way to describe the methods, and further elaboration would be unnecessary considering it is tangential to the main focus of the paper (moral balancing).

R.2.10: The reviewer suggested that the tense in the method was incorrect, particularly in the sentence: “While the majority of the children only speak English,”.

I agree that generally the method should be in past tense. However, in this particular case I am referring to the general demographics of our lab, rather than the demographics of this particular sample. Therefore I feel that present tense is appropriate.

R.2.11: The reviewer pointed out an incorrect capitalisation

This has been corrected.